# Impact of Atezolizumab + Bevacizumab Therapy on Health-Related Quality of Life in Patients with Advanced Hepatocellular Carcinoma

**DOI:** 10.3390/cancers16213610

**Published:** 2024-10-25

**Authors:** Masako Shomura, Haruka Okabe, Maya Sakakibara, Emi Sato, Koichi Shiraishi, Yoshitaka Arase, Kota Tsuruya, Yusuke Mishima, Shunji Hirose, Tatehiro Kagawa

**Affiliations:** 1Faculty of Nursing, Tokai University School of Medicine, 143 Shimokasuya, Isehara-city 259-1193, Kanagawa, Japan; hokabe@tokai.ac.jp (H.O.); maya.73.6292@tokai.ac.jp (M.S.); 2Department of Nursing, University of Tokyo Health Science, 4-11 Ochiai, Tama-city, Tokyo 206-0033, Japan; emi-sat-635@u-ths.ac.jp; 3Division of Gastroenterology, Department of Internal Medicine, Tokai University School of Medicine, 143 Shimokasuya, Isehara-city 259-1193, Kanagawa, Japan; shiraishik@msj.biglobe.ne.jp (K.S.); arase@tokai.ac.jp (Y.A.); ktsuruya@tokai.ac.jp (K.T.); hs9755@tokai.ac.jp (S.H.); kagawa@tokai.ac.jp (T.K.); 4Department of Advanced Medical Science, Tokai University Graduate School of Medicine, 143 Shimokasuya, Isehara-city 259-1193, Kanagawa, Japan; my2987@tokai.ac.jp

**Keywords:** health-related quality of life, hepatocellular carcinoma, immune checkpoint inhibitors, targeted therapy, nursing interventions

## Abstract

This study explored the factors associated with treatment efficacy, treatment duration, and overall survival (OS) in 58 patients with advanced hepatocellular carcinoma undergoing atezolizumab + bevacizumab therapy. Better baseline cognitive and physical function scores and absence of severe (grade ≥ 2) hypoalbuminemia were associated with an improved objective response rate, longer treatment duration, and better OS. These findings highlight the importance of monitoring and managing treatment-related adverse events and maintaining health-related quality of life through multidisciplinary care.

## 1. Introduction

The importance of health-related quality of life (HRQoL) in the management and prognosis of patients with advanced hepatocellular carcinoma (HCC) cannot be overstated. Numerous studies have unequivocally established its significance as a key clinical parameter and a critical research endpoint [1,2]. HRQoL in patients with HCC is influenced by the disease, its complications, treatments, underlying liver disease, and psychological and social aspects [2].

Some studies suggest that liver function may have a stronger association with HRQoL than HCC itself. For example, one study found that the serum albumin level was a better predictor of HRQoL than HCC status [3]. However, another report highlighted the impact of HCC-specific symptoms, such as pain and sleep disorders, on patients’ perceived health status [4].

Moreover, HRQoL has become an essential consideration in the treatment of advanced HCC, influencing clinical decision-making and serving as a prognostic indicator. HRQoL is increasingly being incorporated as an endpoint in clinical trials for new therapies [5,6,7]. The use of validated HRQoL assessment tools, such as the EORTC Core Quality of Life questionnaire and the Functional Assessment of Cancer Therapy–Hepatobiliary questionnaire, allows for comprehensive evaluation of patients’ well-being and can guide treatment strategies to optimize both survival and quality of life [1,2].

Considering the impact of HRQoL is crucial in patients with HCC, as HCC ranks as the third leading cause of cancer-related deaths globally. Understanding and addressing the HRQoL challenges faced by patients with HCC is essential for providing comprehensive care and improving treatment outcomes [8]. It has become a crucial endpoint in evaluating new therapies [9], such as atezolizumab + bevacizumab (Atezo + Bev) [10]. HRQoL can be affected by both cancer symptoms and adverse events (AEs) during treatment. Finn et al. [10] discussed the findings of the IMbrave150 trial, which evaluated Atezo + Bev therapy in patients with unresectable HCC, including its impact on overall survival (OS), progression-free survival, and QoL. Galle et al. [11] further presented the HRQoL analysis from the phase III IMbrave150 trial, focusing on comparing Atezo + Bev and sorafenib as a first-line treatment for unresectable HCC. Similarly, Yau et al. [12] reported phase II results of the IMbrave150 study, focusing on the efficacy, safety, and HRQoL assessment of Atezo + Bev in patients with unresectable HCC who were not eligible for liver transplantation or ablation. Additionally, Bruix et al. [13] reported on the RESORCE trial, which evaluated various aspects, including HRQoL, of regorafenib therapy in patients with HCC whose conditions progressed while receiving sorafenib treatment. Our previous study revealed HRQoL subdomains related to clinical outcomes in patients with advanced HCC [14,15]. A recent study showed that HRQoL is affected by this novel therapy and might contribute to clinical outcomes in patients with advanced HCC treated with Atezo + Bev [16] and pembrolizumab [17]. Collectively, these studies provide comprehensive insights into the assessment of HRQoL in patients with HCC, particularly in the context of novel therapies. Moreover, this analysis may contribute to providing better nursing interventions.

This study, which contributes to this field, aimed to clarify the impact of Atezo + Bev on HRQOL and to identify factors associated with treatment efficacy, treatment duration, and OS in patients with advanced HCC receiving Atezo + Bev by evaluating clinical characteristics, including AEs and HRQoL, at 3 months.

## 2. Materials and Methods

### 2.1. Ethics

This research was implemented according to the Declaration of Helsinki (2000) of the World Medical Association. Ethical approval was obtained from the Institutional Review Board of Tokai University Hospital (NO16R-023). All patients provided written informed consent.

### 2.2. Patients

Consecutive patients with advanced HCC who received Atezo + Bev between 19 November 2020, and 28 December 2023, were included in the study. The patients completed the surveys at treatment initiation (baseline) and monthly after that. They also participated in a nursing intervention program that included education on self-monitoring, AE management, and telephone consultations.

### 2.3. Treatment Procedures

We used the same regimen as in the phase 3 trial of Atezo + Bev [10]. The doses of Atezo and Bev were 1200 mg/body and 15 mg/kg, respectively. Hepatologists decided on dose reduction or discontinuation owing to AEs.

### 2.4. Clinical Evaluation

Treatment efficacy was assessed using the modified Response Evaluation Criteria in Solid Tumors (mRECIST) for 6–9 weeks during the study period [18]. The objective response rate (ORR) was defined as the percentage of patients with complete response (CR) and partial response (PR). The disease control rate (DCR) was the CR, PR, and stable disease (SD) percentage. HRQoL was monitored monthly using the European Organization for Research and Treatment of Cancer Quality of Life Questionnaire Core 30 (EORTC-QLQ C30) [19]. EORTC-QLQ C30 includes general health, five functional subdomains, and nine symptom subdomains. Each subdomain scores 0–100 points. Scores are considered positive if they are higher for general health and the five functional subdomains. However, a higher score is considered negative for the nine symptom subdomains. AEs were assessed using the National Cancer Institute Common Terminology Criteria for Adverse Events version 5.0 [20] at month 3 for landmark analysis. The participants were followed up until 28 December 2023 or death, whichever occurred first.

### 2.5. Statistical Analysis

Relationships between baseline characteristics, AEs, and HRQoL at 3 months with efficacy, OS, and treatment duration were analyzed using multivariate logistic regression models and Cox hazard models with a landmark approach. Event analysis was performed using the Kaplan–Meier method, and statistical significance was determined using the log-rank test. Statistical analyses were performed using IBM SPSS version 26 (IBM Corp., Armonk, NY, USA). The significance level was set at *p* < 0.05.

## 3. Results

### 3.1. Patients’ Baseline Characteristics

A total of 58 patients were enrolled in this study. The study participants were predominantly male (85%), aged ≥ 70 years (45%), and had a Child–Pugh score of 5 (47%) points and Barcelona Clinic Liver Cancer Stage C (41%) (Table 1).

### 3.2. Treatment Efficacy

The efficacy was evaluated in 57 patients. The DCR and ORR were 77.2% and 38.6%, respectively, with a median treatment duration of 11.3 months and a median OS of 20.3 months (Table 2). Thirty-six cases dropped out of Atezo + Bev combination therapy during observation. No patients dropped out because of adverse events. The major cause of dropping out was the progression of disease.

### 3.3. Adverse Events

In total, 52 patients had AEs assessed at 3 months. The most frequently reported AE (all grades) was fatigue (47 cases, 88%), followed by hypoalbuminemia (46 cases, 85%) and thrombocytopenia (35 cases, 63%) (Table 3). There were no grade 4/5 cases in our study. Grade 3 AEs included three cases of proteinuria, two cases of skin toxicity, and one case each of fatigue, thrombocytopenia, anorexia, and diarrhea.

### 3.4. Changes in HRQoL at 3 Months

Notably, HRQoL scores of five functional domains (general health, physical function [PF], role function, emotional function, and cognitive function (CF)) and six symptoms (general fatigue, nausea, pain, dyspnea, insomnia, and financial difficulties) significantly worsened during the first 3 months. There were no significant changes in appetite loss, constipation, or diarrhea (Figure 1).

### 3.5. Factors Associated with ORR

Multivariate analysis revealed that extrahepatic invasion (odds ratio (OR), 0.14; 95% confidence interval (CI), 0.04–0.58; *p* < 0.01) and TNM stage IV (OR, 0.25; 95% CIs, 0.08–0.85; *p* = 0.03) were associated with lower ORRs. Conversely, CF ≥ 80 (OR, 9.10; 95% CIs, 1.78–7.71; *p* < 0.01) at 3 months and grade ≥ 2 skin toxicities (OR, 10.00; 95% CIs, 1.03–100.00; *p* < 0.05) were associated with higher ORRs (Table 4).

### 3.6. Factors Associated with Treatment Duration

Grade ≥2 hypoalbuminemia at 3 months was associated with shorter treatment duration (hazard ratio (HR), 3.39; 95% CI, 1.30–8.84; *p* = 0.01) (Table 5). According to the log-rank analysis, more than grade 2 hypoalbuminemia showed shorter treatment duration than no hypoalbuminemia or grade 1 hypoalbuminemia (Figure 2).

### 3.7. Factors Associated with OS

Age- and sex-adjusted multivariate analysis revealed that patients with baseline des-gamma-carboxy prothrombin (DCP) levels of >1000 mAU/mL had shorter OS than those with lower DCP levels. Meanwhile, grade ≥2 hypoalbuminemia at 3 months was associated with shorter OS (HR, 3.39; 95% CIs, 1.30–8.84; *p* = 0.01). In addition, maintaining a physical function score of ≥80 points at 3 months after the initiation of treatment was essential to achieve a better prognosis (HR, 0.37; 95% CIs, 0.15-0.87; *p* = 0.02) (Table 6). According to the log-rank analysis, more than grade 2 hypoalbuminemia and physical function scores less than 80 showed shorter OS (Figure 3).

## 4. Discussion

This study evaluated the impact of Atezo + Bev therapy on HRQoL of patients with advanced HCC in relation to clinical outcomes. The study population primarily consisted of men (85%), which is consistent with the higher incidence of HCC in this demographic group. Additionally, a significant proportion of patients were older (≥70 years), reflecting the age distribution often observed in patients with HCC. Baseline patient characteristics, including the Child–Pugh score and BCLC stage, indicated a cohort with advanced disease representative of patients typically enrolled in studies evaluating systemic therapies for HCC. Compared with the IMBRAVE 150 cohort [10], our patients were older males with lower liver function and more advanced stages of HCC.

Regarding outcomes, our study showed a DCR of 71.2% and ORR of 38.5%, consistent with the positive findings of previous Atezo + Bev trials for advanced HCC. These results underscore the effectiveness of this combination therapy for achieving disease control and tumor response in this challenging patient population. Moreover, a median treatment duration of 11.3 months and a median OS of 20.3 months demonstrate significant clinical benefits, considering the historically poor prognosis of advanced HCC. Both findings highlight the potential of this combination regimen in improving patient outcomes in HCC. Compared with IMbrave 150, our ORR, DCR, and median OS were almost the same, even if the patients in our study were in more advanced stages of HCC.

In this study, the safety profile of Atezo + Bev was similar to that reported in previous studies, identifying fatigue, hypoalbuminemia, and thrombocytopenia as the most commonly reported AEs. Despite these side effects, most of our patients tolerated the treatment well, as evidenced by the median treatment duration. However, grade ≥ 2 hypoalbuminemia was associated with shorter treatment duration and poorer OS, emphasizing the importance of monitoring and managing treatment-related AEs to optimize outcomes [21]. Specifically, physicians and nurses should evaluate albumin levels before treatment, and nurses and nutritionists should continue to provide preventive nutritional guidance. It is also crucial to prescribe branched-chain amino acids early on, if necessary, depending on albumin levels. Ideal nursing interventions during treatment with Atezo + Bev should include thorough telephone and face-to-face follow-ups. This involves teaching patients to self-monitor and care for their symptoms, as well as evaluating their HRQoL scores to minimize adverse events and improve their HRQoL.

HRQoL assessment is crucial to evaluate the overall impact of therapy on patient well-being. Thus, the significant worsening of HRQoL scores in multiple functional and symptom domains during the first 3 months highlights the need for supportive care interventions to alleviate treatment-related symptoms and improve patient-reported outcomes. Compared with those of a previous study, the HRQOL scores of physical function, role function, and fatigue were worse in patients treated with Atezo + Bev [10], lenvatinib, and sorafenib [14]. Physical function and role function were not detected in patients with any stage of liver cancer [22]; intense nursing care might be effective in improving these functional aspects of HRQOL [23]. Thus, a preferable nursing intervention should be provided to maintain or improve the physical and role functioning of patients with HCC.

In addition, identifying predictive factors associated with treatment response and prognosis, including extrahepatic invasion, TNM stage IV, CF, and skin toxicities, provides valuable insights for patient selection and management. These findings can inform clinical decision-making and help identify those who will benefit the most from Atezo + Bev therapy. According to our results, the predictors of OS with Atezo + Bev are still unclear. Advanced tumor staging and extrahepatic invasion might be predictive factors for clinical outcomes in patients treated with lenvatinib [24] and sorafenib [25]. The tumor staging system is important in observing the clinical outcome [26]. Therefore, we need to compare the predominant staging system in future studies. Grade 2 or 3 skin toxicity may be a preferable predictor of OS in our study. Similarly, a previous study found that mild immune-related AEs might be preferable predictors of OS [27]. Moreover, our previous study showed that grade 2 or 3 skin toxicity may be associated with better clinical outcomes [15]. Grade 2 or 3 hypothyroidism was related to better OS [28]. The original finding was that a better cognitive function score might contribute to longer OS. This is related to a previous study showing that deterioration of cognitive function may cause a greater symptom burden [29]. Our previous study indicated that better social function scores contributed to longer treatment durations [14]. In this study, social functioning was not related to any clinical outcome. However, psychosocial intervention improves social function [30]; therefore, it is essential to assess and support social functioning.

The strengths of our study include a comprehensive analysis of HRQoL in patients with advanced HCC receiving Atezo + Bev combination therapy. We were able to capture changes across multiple domains, such as physical functioning, emotional well-being, social functioning, and cognitive functioning, as well as track changes in symptoms such as pain, fatigue, and nausea. This allowed for a more nuanced understanding of the impact of the treatment on patients’ overall quality of life. The comprehensive evaluation offers important perspectives on how the treatment affects the overall well-being of the patient, going beyond the usual clinical measures. Moreover, by including consecutive patients from a single institution, this study reflects the real-world experiences and challenges faced by patients undergoing Atezo + Bev therapy. This finding adds relevance and applicability to clinical practice, particularly in diverse patient populations. Our study design included monthly surveys, regular HRQoL, AEs, and clinical outcome monitoring. This longitudinal approach allows a nuanced understanding of how treatment affects patients over time and identifies factors associated with treatment efficacy and survival. Additionally, multivariate logistic regression and Cox hazard models were used to analyze factors associated with ORR, treatment duration, and OS, enhancing the robustness of the findings. This rigorous statistical approach helps isolate key outcome predictors and minimizes the impact of confounding variables. Lastly, we compared our results with findings from the IMbrave150 trial and other relevant studies, providing context and validating the observed outcomes against established benchmarks. This comparison strengthened the credibility of the findings of this study.

Nevertheless, our study has some limitations. First, it included 58 patients, which, while sufficient for exploratory analyses, may limit the generalizability of the findings to more extensive or diverse populations. A larger sample size may provide more robust conclusions and enhance the external validity of the results. Second, this study’s findings may be influenced by institutional practices and patient demographics that are unique to this setting. Multicenter studies offer a broader perspective and validate findings across different healthcare environments. Third, the follow-up period until 28 December 2023 provides a snapshot of treatment outcomes and HRQoL changes but may not capture the long-term effects of Atezo + Bev therapy. Extended follow-up is necessary to assess the durability of treatment benefits and long-term HRQoL impacts. Fourth, the inclusion of patients who completed the surveys and received nursing interventions might have introduced a selection bias. Patients who did not adhere to the study protocol or withdrew early may have had different characteristics and outcomes, affecting the overall findings. Fifth, although this study identified several factors associated with treatment response and prognosis, the predictive value of some variables remains unclear. Future research should explore additional biomarkers and patient characteristics to refine treatment efficacy and survival predictors. Finally, we focused solely on Atezo + Bev therapy and did not directly compare its effects with those of other treatment options for advanced HCC. Comparative studies with other therapies may provide a more comprehensive understanding of the relative benefits and limitations of Atezo + Bev.

## 5. Conclusions

This study demonstrated that a significant decline in HRQoL, particularly CF and PF, 3 months after Atezo + Bev therapy initiation in patients with advanced HCC was associated with poor prognosis. Therefore, maintaining HRQoL with appropriate interventions for AEs through a multidisciplinary team approach could contribute to a better prognosis. Although our study findings contribute to the growing body of evidence supporting the efficacy, safety, and impact of this combined regimen, further studies are warranted to validate these findings in larger cohorts and explore strategies to optimize treatment outcomes and patient well-being.

## Figures and Tables

**Figure 1 cancers-16-03610-f001:**
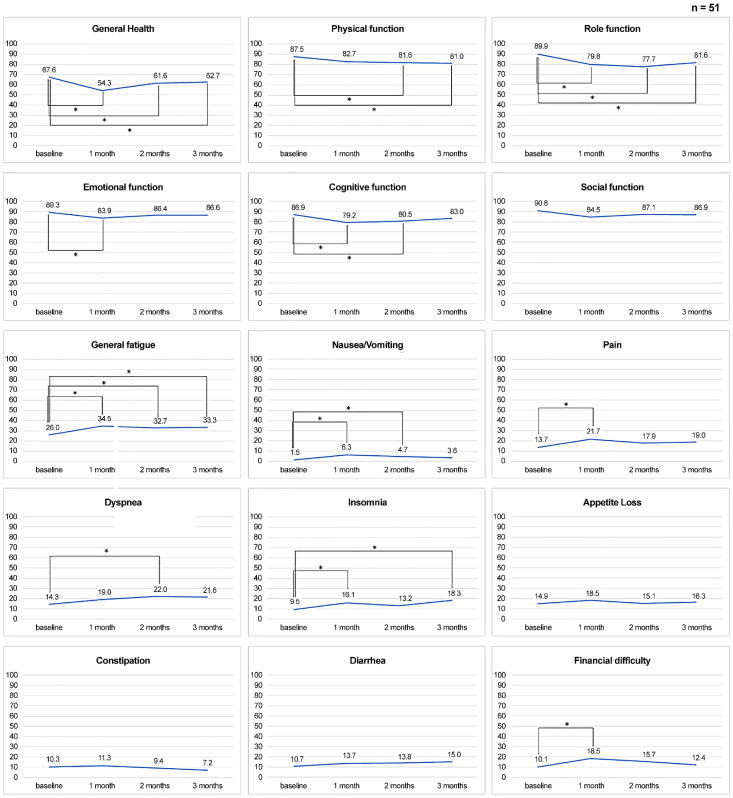
Changes in HRQoL at 3 months. * Asterisks indicate *p*-values less than 0.05.

**Figure 2 cancers-16-03610-f002:**
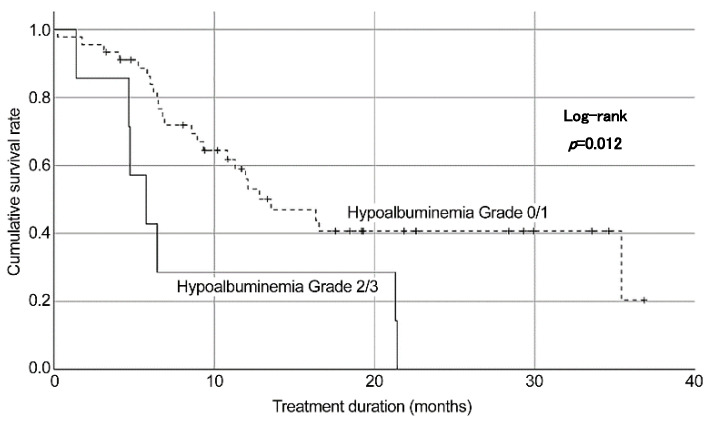
Factors associated with treatment duration.

**Figure 3 cancers-16-03610-f003:**
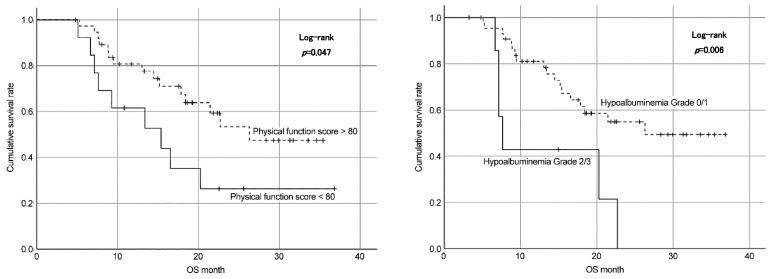
Factors associated with OS.

**Table 1 cancers-16-03610-t001:** Patients’ baseline characteristics (n = 58).

Variables		Number of Cases (%)
Sex	Male Female	49 (85) 9 (15)
Age, y	≧70 <70	26 (45) 32 (55)
BMI	≧22 <22	42 (72) 16 (28)
Etiology	HCV HBV Alcohol NASH Others	20 (34) 10 (17) 13 (22) 8 (14) 7 (12)
Child–Pugh score	5 6 7 8	27 (47) 16 (28) 10 (17) 5 (9)
mALBI grade	1 2a 2b 3	19 (33) 10 (17) 27 (47) 2 (3)
TNM stage	III IV	25 (43) 33 (57)
BCLC (Kindai criteria)	B1 B2 C	10 (17) 24 (41) 24 (41)
Tumor size, mm	≦40 >40	27 (47) 31 (53)
Extrahepatic invasion	Yes No	21 (36) 37 (64)
Vascular invasion	Yes No	9 (16) 49 (84)
History of resection	Yes No	18 (31) 40 (69)
History of TACE	Yes No	34 (59) 24 (41)
History of systemic therapy	Yes No	11 (19) 47 (81)
History of hypertension	Yes No	32 (55) 26 (45)
AFP, ng/mL	>100 ≦100	25 (51) 33 (49)
DCP, mAU/mL	>1000 ≦1000	24 (41) 34 (59)

AFP, alpha-fetoprotein; BCLC, Barcelona Clinic Liver Cancer; BMI, body mass index; DCP, des-gamma-carboxy prothrombin; HBV, hepatitis B virus; HCV, hepatitis C virus; mALBI, modified albumin bilirubin grade; NASH, non-alcoholic steatohepatitis; TACE, transcatheter arterial chemo-embolization.

**Table 2 cancers-16-03610-t002:** Treatment efficacy (n = 57).

Variables		
DCR	77.2%	
ORR	38.6%	CR 2: 3.5% PR 20: 35.1% SD 22: 38.5%
Median treatment duration	11.3 months	
Median OS	20.3 months	

CR, complete response; DCR, disease control rate; ORR, objective response rate; OS, overall survival; PR, partial response; SD, stable disease.

**Table 3 cancers-16-03610-t003:** Occurrence of AEs at 3 months (n = 52).

AE	Grade 1	Grade 2	Grade 3	All Grades
Fatigue	30	16	1	47
Hypoalbuminemia	39	7	0	46
Thrombocytopenia	32	2	1	35
Anemia	27	1	0	28
Skin toxicity	15	6	2	23
Proteinuria	11	7	3	21
Anorexia	10	4	1	15
Hypothyroidism	12	1	0	13
Abdominal pain	11	2	0	13
Dysgeusia	9	0	0	9
Diarrhea	3	3	1	7
Nausea/Vomiting	4	1	0	5
Oral mucositis	5	0	0	5
AST elevation	4	0	0	4
ALT elevation	4	0	0	4
ɤ-GT elevation	4	0	0	4

ɤ-GT, gamma-glutamyl transferase; AE, adverse event; ALT, alanine aminotransferase; AST, aspartate aminotransferase.

**Table 4 cancers-16-03610-t004:** Baseline demographic and clinical variables, AEs, and HRQoL scores associated with ORR (n = 58).

Variables	Univariate	Multivariate
	Crude Odds Ratio (95% CI)	Age- and Sex-Adjusted Odds Ratio (95% CI)
Baseline characteristics		
Men (vs. women)	0.76 (0.17–3.43), *p* = 0.72	
Age, years ≥ 70 (vs. <70)	2.08 (0.70–6.21), *p* = 0.19	
BMI ≥ 22 (vs. <22) kg/m^2^	2.35 (0.65–8.52), *p* = 0.19	
HCV infection (vs. other etiology)	0.66 (0.22–2.01), *p* = 0.47	
Child-Pugh Score 5 (vs. over 6)	1.60 (0.55–4.68), *p* = 0.39	
mALBI grade = 1/2a (vs. 2b/3)	1.72 (0.58–5.05), *p* = 0.33	
TNM stage III (vs. IV)	3.15 (1.04–9.56), *p* = 0.036	3.89 (1.16–13.12), *p* = 0.028
BCLC B1/B2 (vs. C)	0.82 (0.25–2.66), *p* = 0.74	
Maximum tumor size, >40 mm (vs. <40 mm)	1.33 (0.46–3.89), *p* = 0.60	
Extrahepatic invasion − (vs. +)	6.71 (1.68–26.83), *p* = 0.007	6.26 (1.53–25.59), *p* = 0.011
Vascular invasion − (vs. +)	1.31 (0.29–5.88), *p* = 0.72	
History of resection + (vs. −)	0.43 (0.13–1.34), *p* = 0.15	
History of TACE + (vs. −)	1.25 (0.43–3.67), *p* = 0.69	
History of systemic therapy + (vs. −)	3.46 (0.67–17.83), *p* = 0.14	
History of hypertension + (vs. −)	0.54 (0.18–1.61), *p* = 0.27	
AFP, ng/mL ≥100 (vs. <100)	1.65 (0.55–4.93), *p* = 0.37	
DCP, mAU/mL ≥1000 (vs. <1000)	1.08 (0.37–3.20), *p* = 0.89	
AEs in 3 months		
Hypoalbuminemia grade 2/3 (vs. grade 0/1)	NE (NE), *p* = 0.99	
Skin toxicity grade 2/3 (vs. grade 0/1)	12.5 (1.47–100.00), *p* = 0.021	16.6 (1.71–166.67), *p* = 0.016
HRQoL score at 3 months		
Physical function ≥ 80 (vs. <80)	3.33 (0.79–14.05), *p* = 0.10	3.26 (0.75–14.29), *p* = 0.12
Cognitive function ≥ 80 (vs. <80)	9.33 (1.84–47.44), *p* = 0.007	9.13 (1.74–47.81), *p* = 0.009

CI, confidence interval; HRQoL, health-related quality of life; NE, not estimated.

**Table 5 cancers-16-03610-t005:** Baseline demographic and clinical variables, AEs, and HRQoL scores associated with treatment duration (n = 58).

Variables	Univariate	Multivariate
	Crude HR (95% CI)	Age- and Sex-Adjusted HR (95% CI)
Baseline characteristics		
Men (vs. women)	1.36 (0.56–3.29), *p* = 0.50	
Age, years > 70 (vs. <70)	1.05 (0.55–2.02), *p* = 0.89	
BMI < 22 (vs. >22) kg/m^2^	0.85 (0.41–1.77), *p* = 0.66	
HCV infection (vs. other etiology)	0.85 (0.43–1.69), *p* = 0.64	
Child-Pugh Score 5 (vs. over 6)	1.31 (0.67–2.56), *p* = 0.43	
mALBI grade = 1/2a (vs. 2b/3)	1.25 (0.65–2.42). *p* = 0.51	
TNM stage III (vs. IV)	0.96 (0.49–1.87), *p* = 0.90	
BCLC B1/B2 (vs. C)	1.38 (0.65–2.97), *p* = 0.41	
Maximum tumor size, >40 mm (vs. <40 mm)	1.26 (0.65–2.47), *p* = 0.49	
Extrahepatic invasion − (vs. +)	0.78 (0.39–1.58), *p* = 0.50	
Vascular invasion − (vs. +)	0.54 (0.19–1.55), *p* = 0.25	
History of resection + (vs. −)	1.20 (0.60–2.42), *p* = 0.60	
History of TACE + (vs. −)	1.22 (0.62–2.38), *p* = 0.57	
History of systemic therapy + (vs. −)	1.35 (0.63–2.89), *p* = 0.44	
History of hypertension + (vs. −)	0.94 (0.49–1.83), *p* = 0.86	
AFP, ng/mL >100 (vs. <100)	1.05 (0.54–2.03), *p* = 0.90	
DCP, mAU/mL >1000 (vs. <1000)	1.55 (0.79–3.07), *p* = 0.20	
AEs in 3 months		
Hypoalbuminemia grade 2/3 (vs. grade 0/1)	2.85 (1.21–6.69), *p* = 0.016	3.05 (1.24–7.51), *p* = 0.015
Skin toxicity grade 2/3 (vs. grade 0/1)	1.53 (0.62–3.76), *p* = 0.36	
HRQoL score at 3 months		
Physical function > 80 (vs. <80)	1.52 (0.71–3.25), *p* = 0.28	
Cognitive function > 80 (vs. <80)	1.23 (0.58–2.58), *p* = 0.59	

HR, hazard ratio.

**Table 6 cancers-16-03610-t006:** Baseline demographic and clinical variables, AEs, and HRQoL scores associated with OS (n = 58).

Variables	Univariate	Multivariate
	Crude HR (95%CI)	Age- and Sex-Adjusted HR (95%CI)
Baseline characteristics		
Men (vs. women)	2.17 (0.65–7.21), *p* = 0.21	
Age, years > 70 (vs. <70)	0.65 (0.31–1.36), *p* = 0.25	
BMI < 22 (vs. >22) kg/m^2^	1.45 (0.68–3.13), *p* = 0.34	
HCV infection (vs. other etiology)	1.09 (0.52–2.31), *p* = 0.82	
Child–Pugh Score 5 (vs. over 6)	1.61 (0.76–3.42), *p* = 0.21	
mALBI grade = 1/2a (vs. 2b/3)	1.44 (0.69–3.00), *p* = 0.33	
TNM stage III (vs. IV)	0.92 (0.44–1.90), *p* = 0.81	
BCLC B1/B2 (vs. C)	1.47 (0.65–3.34), *p* = 0.36	
Maximum tumor size, >40 mm (vs. <40 mm)	1.22 (0.58–2.54), *p* = 0.60	
Extrahepatic invasion − (vs. +)	0.53 (0.24–1.21), *p* = 0.13	
Vascular invasion − (vs. +)	0.66 (0.20–2.20), *p* = 0.50	
History of resection	1.36 (0.63–2.93), *p* = 0.44	
History of TACE	1.03 (0.44–2.19), *p* = 0.93	
History of systemic therapy	1.41 (0.61–3.32), *p* = 0.42	
History of hypertension	1.12 (0.54–2.33), *p* = 0.76	
AFP, ng/mL >100 (vs. <100)	1.06 (0.51–2.20), *p* = 0.88	
DCP, mAU/mL >1000 (vs. <1000)	2.36 (1.10–5.05), *p* = 0.027	2.29 (1.06–4.95), *p* = 0.036
AEs at 3 months		
Hypoalbuminemia grade 2/3 (vs. grade 0/1)	3.43 (1.34–8.76), *p* = 0.010	3.39 (1.30–8.84), *p* = 0.013
Skin toxicity grade 2/3 (vs. grade 0/1)	1.65 (0.61–4.47), *p* = 0.321	
Physical function > 80 (vs. <80)	0.44 (0.20–1.01), *p* = 0.053	0.36 (0.15–0.86), *p* = 0.021
Cognitive function > 80 (vs. <80)	0.75 (0.32–1.69), *p* = 0.48	

## Data Availability

Data supporting this study’s findings are available from the corresponding author upon reasonable request.

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
