# Peer review of "Impact of Atezolizumab + Bevacizumab Therapy on Health-Related Quality of Life in Patients with Advanced Hepatocellular Carcinoma"

_cancers, 2024, doi:10.3390/cancers16213610_

Round 1
Reviewer 1 Report
Comments and Suggestions for Authors
The authors present the outcomes from a retrospective single center study aiming to explore the factors associated with treatment efficacy, treatment duration, and overall survival in patients with advanced HCC undergoing Atezo+Beva therapy.
This is an excellent study focusing on a very important aspect of this particular patient group despite the limited number of patients included.
The title reflects the subject of the manuscript. It presents a clear and clinically useful message. It is well written in terms of clarity, style, and use of English language. The discussion section is sufficiently detailed and explains adequately the purpose of this study in the context of published information. The conclusion accurately and clearly explains the main result. The length of the manuscript is ideal. All references are appropriate and current.
Minor point: L193 - need to add a reference(s) for previous trials
Author Response
Reviewer 1
Comments: The authors present the outcomes from a retrospective single center study aiming to explore the factors associated with treatment efficacy, treatment duration, and overall survival in patients with advanced HCC undergoing Atezo+Beva therapy.
This is an excellent study focusing on a very important aspect of this particular patient group despite the limited number of patients included.
The title reflects the subject of the manuscript. It presents a clear and clinically useful message. It is well written in terms of clarity, style, and use of English language. The discussion section is sufficiently detailed and explains adequately the purpose of this study in the context of published information. The conclusion accurately and clearly explains the main result. The length of the manuscript is ideal. All references are appropriate and current.
#1 Minor point: L193 - need to add a reference(s) for previous trials
- Answer: Thank you for your suggestion. I added the next reference to this part, “10. Finn, R.S.; Qin, S.; Ikeda, M.; Galle, P.R.; Ducreux, M.; Kim, T.Y.; Kudo, M.; Breder, V.; Merle, P.; Kaseb, A.O.; et al. Atezolizumab plus bevacizumab in unresectable hepatocellular carcinoma. N Engl J Med 2020, 382, 1894–1905. DOI:10.1056/NEJMoa1915745.” The revised sentence is below.
“Regarding outcomes, our study showed a DCR of 71.2% and ORR of 38.5%, consistent with the positive findings of previous Atezo+Bev trials for advanced HCC [10].”
Reviewer 2 Report
Comments and Suggestions for Authors
This manuscript investigates the effects of atezolizumab and bevacizumab therapy on the quality of life and treatment efficacy in patients with advanced HCC. Although the results are hampered by the small sample size, I think that they may have some relevance in the current scientific debate in the field.
Some suggestions:
- can the Authors expand on the rationale for conducting this study?
- similarly, the Authors did a great job in addressing the limitations of the study, but the actual relevance of their findings should be better clarified.
- the Introduction should be more focused on the unmet need investigated in the study, and be less wordy on general background.
- quality of figures must be greatly improved.
- please focus on references published in the last 3 years
Comments on the Quality of English LanguageModerate polishing required
Author Response
Reviewer 2
Comments: This manuscript investigates the effects of atezolizumab and bevacizumab therapy on the quality of life and treatment efficacy in patients with advanced HCC. Although the results are hampered by the small sample size, I think that they may have some relevance in the current scientific debate in the field.
Some suggestions:
#1 Can the Authors expand on the rationale for conducting this study?
- Answer: Thank you for your suggestion. I added the following descriptions to the introduction L63–L66:
“Considering the impact of HRQoL is crucial in patients with HCC, as HCC ranks as the third leading cause of cancer-related deaths globally. Understanding and addressing the HRQoL challenges faced by HCC patients is essential for providing comprehensive care and improving treatment outcomes [8].”
#2 Similarly, the Authors did a great job in addressing the limitations of the study, but the actual relevance of their findings should be better clarified.
- Answer: Thank you for your suggestion. I added the following strength of this study to L259–L266:
“The strengths of our study include a comprehensive analysis of HRQoL in patients with advanced HCC receiving Atezo+Bev combination therapy. We were able to capture changes across multiple domains such as physical functioning, emotional well-being, social functioning, and cognitive functioning, as well as track changes in symptoms such as pain, fatigue, and nausea. This allowed for a more nuanced understanding of the impact of the treatment on patients' overall quality of life. The comprehensive evaluation offers important perspectives on how the treatment affects the overall well-being of the patient, going beyond the usual clinical measures.”
#3 The Introduction should be more focused on the unmet need investigated in the study, and be less wordy on general background.
- Answer: Thank you for your suggestions. I tried to improve the Introduction by focusing more on the unmet needs of investigating our study and making it less wordy on the general background.
L45–L53 was revised as follows: “The importance of health-related quality of life (HRQoL) in the management and prognosis of patients with advanced hepatocellular carcinoma (HCC) cannot be overstated. Numerous studies have unequivocally established its significance as a key clinical parameter and a critical research endpoint [1, 2]. HRQoL in HCC patients is influenced by the disease, its complications, treatments, underlying liver disease, and psychological and social aspects [2]. Some studies suggest that liver function may have a stronger association with HRQoL than HCC itself. For example, one study found that serum albumin level was a better predictor of HRQoL than HCC status [3].”
Also, the following sentence written in red was added to L82–L85: “Collectively, these studies provide comprehensive insights into the assessment of HRQoL in patients with HCC, particularly in the context of novel therapies. Moreover, this analysis may contribute to providing better nursing interventions.”
#4 Quality of figures must be greatly improved.
- Answers: Thank you for your suggestion. I tried to improve the readability of all the figures. Moreover, I added a p-value in each figure if necessary.
#5 Please focus on references published in the last 3 years
- Answer: Thank you for your suggestion. I rechecked our list of references thoroughly. As a result, adequate references from more recent years could not be found for this manuscript. Therefore, I could not change any references to more recent ones.
Reviewer 3 Report
Comments and Suggestions for Authors
Reviewer’s comments
This original article written by Shomura et al. primarily focuses on the usefulness of care intervention rather than treatment outcomes in HCC patients who underwent the treatment with Atezo + Bev. This original article has its own originality very well and seems to be very interesting. However, the explanation for the methods in this study is insufficient or ambiguous in several parts. The interpretation of the results obtained from this study is also insufficient. Several revisions are required to improve the quality of this paper. Please refer to the comments shown below.
Major
#1. Keywords reflecting this study were not selected. Another keyword indicating care intervention should be added.
#2. The authors did not describe the aim of this study very well. Another aim of this study was to verify the clinical significance of care intervention in HCC patients who underwent Atezo+Bev treatment. The authors should clearly note that in “Introduction”.
#3. The explanation for HRQoL score was missing. For example, how was the score calculated in each parameter? How were the role function, emotional function, and cognitive function evaluated?
#4. In Table 1, items for local treatments including radiofrequency ablation and percutaneous ethanol injection treatment were missing as past treatment history for HCC.
#5. In Table 2, why was the treatment efficacy not evaluated in one patient (n=57) ? The authors should describe the reason.
#6. How many patients have dropped out of this treatment during observation period. It is one of important parameters. Have some patients dropped out because of severe adverse events?
#7. In Figure 1, role function and cognitive function were restored at 2 or 3 months. These findings seem to be interesting. The author should describe or speculate the reasons.
#8. In Table 4, why was hypoalbuminemia not estimated? Moreover, hypoalbuminemia is significantly associated with treatment duration and OS, as shown in Table 5 and Table 6. Hypoalbuminemia indicates both malnutrition and unfavorable hepatic reserve. Which factor was related to hypoalbuminemia in this study.
#9. This study highlighted the need for care intervention in HCC patients who underwent Atezo+Bev treatment. However, the content of care intervention in such patients was not described in detail. What did supportive care intervention and intense nursing care indicate (Page 10, lines 213 and 218). What should nurses do as the nursing care?
#10. The statements on Figure 2 and Figure 3 were missing. A p-value should be also noted in each figure.
Minor
#1. Page 3, line 100, Page 9, line 207: The term, “physicians” is not appropriate. The term should be replaced with hepatologists.
#2. Page 3, lines 99-100: The statement, “The patients received Atezo+Bev intravenously at 1200 mg/body and 15mg/kg”, is strange. The sentence should be corrected to “The doses of Atezo and Bev were 1200mg/body and 15mg/kg, respectively”.
#3. Table 1: “Alcoholic” should be corrected to” Alcohol”.
#4. Page 10, line 227-228: The term, “tumor staging method” is not appropriate. The term should be replaced with “tumor staging system”.
#5. What did asterisks indicate in Figure 1? Did they mean p-values less than 0.05?
Comments on the Quality of English Languageits quality is quite excellent.
Author Response
Reviewer 3
Comments: This original article written by Shomura et al. primarily focuses on the usefulness of care intervention rather than treatment outcomes in HCC patients who underwent the treatment with Atezo + Bev. This original article has its own originality very well and seems to be very interesting. However, the explanation for the methods in this study is insufficient or ambiguous in several parts. The interpretation of the results obtained from this study is also insufficient. Several revisions are required to improve the quality of this paper. Please refer to the comments shown below.
Major
#1. Keywords reflecting this study were not selected. Another keyword indicating care intervention should be added.
- Answer: Thank you for your suggestions. I added “nursing interventions” as a keyword of this study.
#2. The authors did not describe the aim of this study very well. Another aim of this study was to verify the clinical significance of care intervention in HCC patients who underwent Atezo+Bev treatment. The authors should clearly note that in “Introduction”.
- Answer: Thank you for your suggestions. I improved and added a clearer expression for the purpose of the study at the end of the Introduction as follows (L86–L89):
“This study, which contributes to this field, aimed to clarify the impact of Atezo+Bev on HRQOL and to identify factors associated with treatment efficacy, treatment duration, and OS in patients with advanced HCC receiving Atezo+Bev based on evaluating clinical characteristics, including AEs and HRQoL, at 3 months.”
#3. The explanation for HRQoL score was missing. For example, how was the score calculated in each parameter? How were the role function, emotional function, and cognitive function evaluated?
- Answer: Thank you for your suggestions. I added the explanation and interpretation of HRQoL scores as follows (L113–L116):
“…(EORTC-QLQ C30). EORTC-QLQ C30 includes general health, five functional subdomains, and nine symptom subdomains. Each subdomain scores 0–100 points. Scores mean positive if higher in general health and five subdomains. However, a higher score means negatively in nine symptom subdomains.”
#4. In Table 1, items for local treatments including radiofrequency ablation and percutaneous ethanol injection treatment were missing as past treatment history for HCC.
- Answer: Thank you for your suggestions. In our patient group, the number of patients who underwent radiofrequency ablation and percutaneous ethanol injection treatment was almost zero; therefore, we did not include two of these measures as past treatment history.
#5. In Table 2, why was the treatment efficacy not evaluated in one patient (n=57) ? The authors should describe the reason.
- Answer: Thank you for your suggestions. Due to the initiation timing, one patient did not reach the evaluation of efficacy from 6 weeks to 9 weeks. Because of this, we lacked one patient’s efficacy data.
#6. How many patients have dropped out of this treatment during observation period. It is one of important parameters. Have some patients dropped out because of severe adverse events?
- Answer: Thank you for your suggestions. I added the following sentences to L140–L141:
“Thirty-six cases dropped out of Atezo+Bev combination therapy during observation. No patients dropped out because of adverse events.”
#7. In Figure 1, role function and cognitive function were restored at 2 or 3 months. These findings seem to be interesting. The author should describe or speculate the reasons.
- Answer: Thank you for your suggestions. It might be considered that most patients get used to managing their role in their home or social environment after more than two months of therapy. Adjusting their condition with therapy makes the patients feel cognitively fine. The previous study did not show comparative results; therefore, I cannot add this opinion to our manuscript.
#8. In Table 4, why was hypoalbuminemia not estimated? Moreover, hypoalbuminemia is significantly associated with treatment duration and OS, as shown in Table 5 and Table 6. Hypoalbuminemia indicates both malnutrition and unfavorable hepatic reserve. Which factor was related to hypoalbuminemia in this study.
- Answer: Thank you for your suggestions. We did not observe severe anorexia due to Atezo+Bev treatment; therefore, before the research observation, we could not estimate hypoalbuminemia. We could not find out why we had observed hypoalbuminemia, which caused a shorter treatment duration and OS. We are considering using hypoalbuminemia as our future research task.
#9. This study highlighted the need for care intervention in HCC patients who underwent Atezo+Bev treatment. However, the content of care intervention in such patients was not described in detail. What did supportive care intervention and intense nursing care indicate (Page 10, lines 213 and 218). What should nurses do as the nursing care?
- Answer: Thank you for your su I added ideal nursing interventions, which were suggested in this study, in L226–L229, as follows:
“Ideal nursing interventions during treatment with Atezo+Bev should include thorough telephone and face-to-face follow-ups. This involves teaching patients to self-monitor and care for their symptoms, as well as evaluating their HRQoL scores to minimize adverse events and improve their HRQoL.”
#10. The statements on Figure 2 and Figure 3 were missing. A p-value should be also noted in each figure.
- Answer: Thank you for your suggestion. I added descriptions of Figures 2 and 3 to our manuscripts. Moreover, a p-value was added to each figure.
Minor
#1. Page 3, line 100, Page 9, line 207: The term, “physicians” is not appropriate. The term should be replaced with hepatologists.
- Answer: Thank you for your suggestion. I have revised “physician” to “hepatologist” in L104.
#2. Page 3, lines 99-100: The statement, “The patients received Atezo+Bev intravenously at 1200 mg/body and 15mg/kg”, is strange. The sentence should be corrected to “The doses of Atezo and Bev were 1200mg/body and 15mg/kg, respectively”.
- Answer: Thank you for your very kind suggestions. I revised in L103–L104 as follows:
“The doses of Atezo and Bev were 1200 mg/body and 15 mg/kg, respectively.”
#3. Table 1: “Alcoholic” should be corrected to” Alcohol”.
- Answer: Thank you for your suggestion. I revised “Alcoholic” to “Alcohol” in Table 1.
#4. Page 10, line 227-228: The term, “tumor staging method” is not appropriate. The term should be replaced with “tumor staging system”.
- Answer: Thank you for your suggestions. I revised “tumor staging method” to “tumor staging system” in L246–L248 as follows:
“The tumor staging system is important in observing the clinical outcome [26]. Therefore, we need to compare the predominant staging system in future studies.”
#5. What did asterisks indicate in Figure 1? Did they mean p-values less than 0.05?
- Answer: Thank you for your suggestion. The asterisks in Figure 1 show p-values less than 0.05. I added the note* below Figure 1.
Reviewer 4 Report
Comments and Suggestions for Authors
Shomura et al offered an interesting paper, reporting experience in HCC immunotherpay with ATZ-BEVA, aiming quality of life assessment. Despite a low originalty, the article is well-wirtten and globally interesting.
I want to point HRQOL. Do the authors see other scores: QLQ-HCC18, FACT-HEP ?
Author Response
Reviewer 4
Shomura et al offered an interesting paper, reporting experience in HCC immunotherpay with ATZ-BEVA, aiming quality of life assessment. Despite a low originalty, the article is well-wirtten and globally interesting.
#1 I want to point HRQOL. Do the authors see other scores: QLQ-HCC18, FACT-HEP?
- Answer: Thank you for your suggestions. At IMBRAVE 150, phase 3 study of Atezo+Bev, they used EORTC-QLQ C30 and HCC18. Therefore, we followed IMBRAVE 150 to choose EORTC-QLQ C30 and HCC18 for their comparativeness, not FACT-HEP. We also evaluated EORTC QLQ-HCC18. For this manuscript, we did not use data from QLQ-HCC18 because it did not show a relationship between clinical outcomes. Moreover, using those data would complicate the results.
Round 2
Reviewer 2 Report
Comments and Suggestions for Authors
thanks for having addressed my comments
Reviewer 3 Report
Comments and Suggestions for Authors
Reviewer’s comments to the revised manuscript
Major
#4. According to the reply from the authors, almost none of the enrolled HCC patients underwent local treatment as past treatment for HCC, although 18 and 34 patients did hepatic resection and TACE, respectively. It seems to be quite strange. In addition, the term, “almost zero” is quite ambiguous. If the statement is true, that should be noted in Table 1.
#7. The authors did not respond to the reviewer’s comment #7 very well. The recovery of role function and cognitive function at 2,3 months may be derived from nursing care intervention. The novel evidence seems to be very interesting. However, the authors did not mention the reason in detail. This study has been conducted with only one arm (patients with nursing care). Therefore, another arm (patients without nursing care) should have been added to confirm the efficacy of nursing care intervention as a study design.
#8. In Table 4, hypoalbuminemia was N/E(not estimated), although p-value was 0.99. The reason remains unclear. The authors focused on hypoalbuminemia as an adverse event (AE) by the treatment with Atezo+Bev in the enrolled HCC patients. Hypoalbuminemia as the AE is usually derived from severe proteinuria. However, hypoalbuminemia is also derived from malnutrition or unfavorable hepatic reserve. In this article, the cause of hypoalbuminemia in such HCC patients has not been fully mentioned. Was hypoalbuminemia caused by an AE? Or was that caused by the disease progression? Did the authors check severe proteinuria as the cause of hypoalbuminemia in those patients? May the discontinuation of the treatment with Atezo+Bev due to the AE account for unfavorable overall survival in this study?
The efficacy of branched-chain amino acid (BCAA) was described in the text. However, the prescription is basically considered for hypoalbuminemia due to unfavorable hepatic reserve (not proteinuria). The authors may confuse the cause of hypoalbuminemia in those HCC patients.
Comments on the Quality of English LanguageMinor revision is required in this manuscript.